# Immersion Freezing of a Scots Pine Single Seed in a Water-Saturated Dispersion Medium: Mathematical Modelling

**Olga Dornyak and Arthur Novikov ***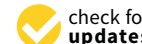

Mechanical Department, Voronezh State University of Forestry and Technologies Named after G.F. Morozov, 8, Timiryazeva, 394087 Voronezh, Russia; ordornyak@mail.ru

*   Correspondence: arthur.novikov@vglta.vrn.ru; Tel.: +7-903-650-84-09

**Abstract:** Forest owners will be able to solve the problem of protecting small forest seeds from mechanical and atmospheric influences during aerial sowing, as well as the problem of manufacturing capsules in the field, saving financial, time and material resources. The process of creating a capsule by freezing the seed in a water-saturated dispersed system—immersion freezing—allows you to organize the technological properties of forest seeds depending on the initial requirements. In most cases, the quality of the seed capsule is determined by the thermophysical and mechanical properties of the components. The technological process of obtaining seed capsules for aerial seeding and the choice of freezing modes is based on a priori mathematical modeling of heat-and-mass transfer processes. The main purpose of the study is to predict the duration of the seed freezing process in a capsule with a water-saturated dispersed medium, depending on the external temperature conditions, the geometric parameters of the capsule and the seed. The cooling agent is carbon dioxide. The research is based on the use of numerical modeling methods on the platform COMSOL Multiphysics. A mathematical model is proposed that allows us to obtain the dynamics of the distribution of temperature and moisture content fields in the dispersed system and seed depending on a complex of geometric and thermophysical factors. The time of immersion freezing of the capsule with the common pine seed for the conditions considered should be in the range of 150 to 250 s.

**Keywords:** small-size forest seeds; *Pinus sylvestris* L.; liquid immersion freezing; heat-and-mass transfer; seed grading; optoelectronic grader; numerical modeling

## 1. Introduction

Automated aerial seeding [1–3] of small-size seeds in forest landscape restoration [4–7] using Unmanned Aerial Vehicles (UAV) [3,8,9] imposes the following requirements on the seed material:

*   uniformity of seeds morphometric characteristics and their placement in seed capsules, providing increased accuracy of seeding, since without placing seeds in capsules, according to Masarei et al. [10], due to the heterogeneity of the size and shape of seeds, the depth of their sealing is significantly different even from ground-based seeding;
*   aerodynamic stability and strength of the seed capsule, and the ability to manufacture it in the field [11].
*   immutable initial seed germination potential or possible additional activation of growth processes and increased nutrient supply.

To implement these requirements, the technology of seed preparation is proposed, which involves their immersion in a water-saturated dispersed medium based on river sand ($SiO_2$—98%, the rest $Al_2O_3$, and $Fe_2O_3$) and subsequent freezing in a capsule of the desired shape.

It is expected that when the water freezes, a capsule with a stronger structural frame will be formed compared to a pure ice shell. Freezing in the field can be carried out by using carbon dioxide pellets as the freezing agent, which does not have a harmful effect on nature and humans. The shape of the capsule should be chosen in accordance with recommendations aimed at better aerodynamic characteristics [12], as well as to ensure more accurate seeding.

It should be noted that changing the aerodynamic properties of seeds is also possible with such methods of their preliminary preparation as seed coating [13] and incrustation [14], but these methods imply the impossibility of use in the field, and significantly higher costs.

When frozen in air by immersion of hermetically packed *Pinus* seeds in liquid nitrogen −196 °C [15] to a temperature of −172 °C, the seeds retain germination without cryoprotectors [16]. At the same time, the quality of cryopreservation of small forest seeds is determined by the parameters of the freezing rate and humidity of seeds [15].

The lack of control over the rate of temperature change during the "freeze-thaw" process leads to a noticeable deterioration in germination indicators; in pine trees, the difference from the control for germination energy is 8–16%, while for laboratory germination it is 10–13% [17]. While 80–90% of the pine seeds there is no cracking of the outer shell (cryoscarification). At the same time, Jabłoński et al. [18] report the need for scarification of seeds before seeding with a sufficiently hard shell (including common pine) after long-term storage in order to intensify germination processes. Large forest seeds are in most cases sensitive to drying [19], and small ones, especially common pine, can be dried to a humidity of 4% for better low temperature transfer [15]. Seeds of the *Pinus* species, frozen in air at a temperature of −18 °C and stored further at the same temperature, have advantages in terms of germination indicators over seeds stored at a temperature of +4 °C [20,21].

There are practically no experimental studies related to freezing forest seeds by immersion in a liquid medium. In a review of seed encapsulation strategies from 1949 to 2019, conducted by Cornish et al. [21], there is no information about the possible content of water in the capsules during the freezing stage. There are papers in the field of freezing biological objects by immersion in liquid, the purpose of which is to preserve food products [22–26].

Compared to freezing in the air, immersion freezing is accompanied by more intense heat-and-mass transfer processes both in the seed and in the surrounding shell. At the same time, in biological objects, as shown in Chourott et al. 's work, mass transfer is reduced not only to the transport of water, but also to the diffusion of substances dissolved in water [26]. This process, according to Lucas and Raoltwack, can be controlled by changing the time of immersion freezing» [24]. It is possible to intensify the freezing process, as shown in Sun et al. [27] using ultrasound exposure.

The purpose of this work is to create a mathematical model of heat-and-mass transfer during the cooling and freezing of seeds by immersion in water-saturated sand dispersion to predict the duration of the freezing process depending on external temperature influences and the geometric parameters of the seed capsule and seed. The freezing mode should provide a slight wetting of the seed compared to the initial optimal humidity value, since an increase in the moisture content of the seed reduces its germination potential. In addition, it is advisable not to lower the average seed temperature below −18 °C, since it has been proven that such cooling practically does not affect the germination of pine seeds.

## 2. Materials and Methods

The subject of the study was the seeds of Scots pine (*Pinus sylvestris* L.), the shape of which was approximated by an ellipsoid [28]. Seeds were obtained from cones collected in a natural forest stand of Voronezh region, Russia [29]. Statistical processing of Scots pine seeds size measurements gave values of length *l*, thickness *b* and width *c* of the seed. It was obtained that *l* = 4–6 mm, *b* = 1.5–2 mm, and *c* = 2.5–3 mm (standard deviation = 0.47 mm, N = 105).

The object of the study is the process of immersion freezing of a single seed of Scots pine in the process of creating ice capsules for aerial seeding according to the patent [12], developed with the

participation of the authors. To describe the liquid immersion freezing of a single seed in a water-based conduit, the basic provisions of heat-and-mass transfer in porous media theory were used [30,31]. The mathematical model was studied numerically in a 2D approximation using the universal modeling environment COMSOL Multiphysics.

## 3. Mathematical Model

A form made of cold-resistant material with individual cells (1 in Figure 1a) of a certain configuration is filled to a certain level with sand (2). Then a single dewinged seed (3) is placed on the sand cushion and covered with sand again. The addition of water occurs in the volume that completely saturates the porous space. Thus, the sample to be frozen consists of two materials: a water-saturated system based on sand and an unsaturated colloidal capillary-porous material—the seed. It is assumed that the formation of the billet in this way will ensure a more accurate distribution of sown seeds on the soil surface due to the shape of the capsule and its integrity when falling due to a strong structural framework of frozen sand mixture.

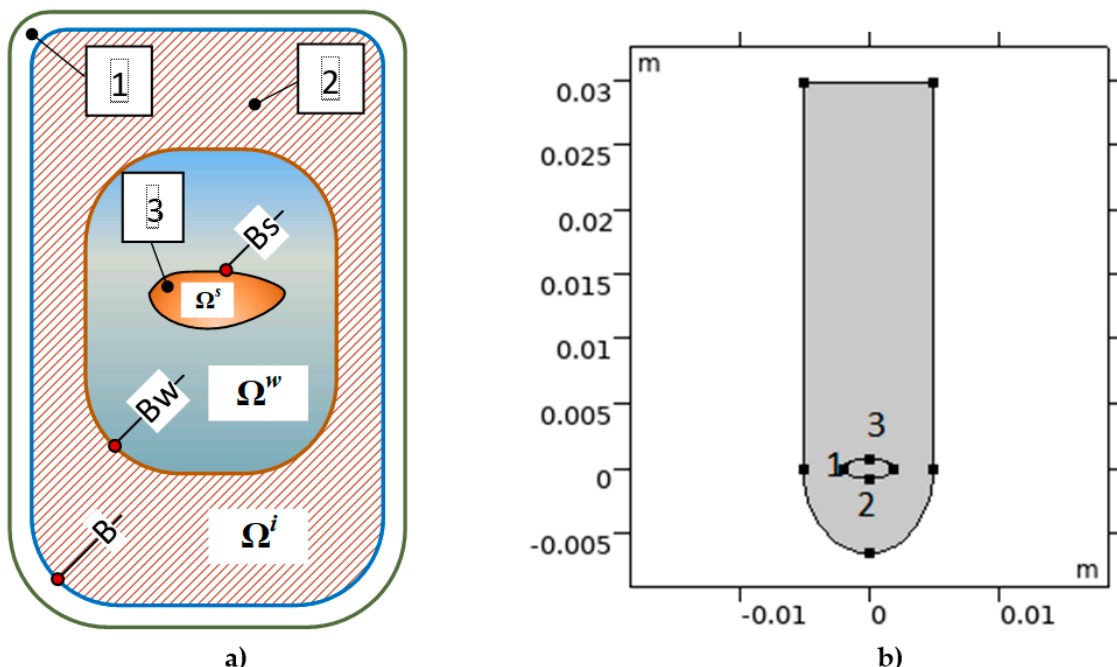

**Figure 1.** Structural (**a**) and computational (**b**) schemes for the model of heat-and-mass transfer in the formation of an ice capsule during aerial seeding. $\Omega^i$ and $\Omega^w$ are zones of a water-saturated sand-based dispersion system containing water in the solid and liquid phases, respectively, and $\Omega^s$-zone is an unsaturated colloidal capillary-porous material—the seed. The letter B with the corresponding index indicates the boundary between phases or media. The index "*s*" corresponds to seed, "*w*" to water, and "*i*" to ice.

The duration of the freezing process is one of the main characteristics of the technology under consideration. The freezing mode should provide a slight wetting of the seed compared to the initial humidity value, since an increase in the moisture content of the seed reduces its germination potential, on the one hand. On the other hand, the freezing process should be energy efficient. Determining the duration of the process depending on the complex of geometric and thermophysical parameters can be performed using mathematical modeling methods. In this paper, a non-stationary mathematical model of the heat-and-mass transfer processes in the sample during freezing is proposed based on the A.V. Lykov's equations [30]. The following assumptions are accepted:

(I). The front of the phase transition of a saturated coarse—dispersed system "sand-water" is a region with a dimension one less than the dimension of the problem, i.e., the interphase boundary is "sharp", not blurred, which is true when the proportion of bound water in a capillary-porous system is insignificant compared to the proportion of free water;

(II). The influence of the temperature gradient on the transfer of moisture in the system can be ignored;

(III). The seed has water only in the bound state.

### 3.1. Heat Task Definition

According to assumptions (I)–(II), the Stefan problem statement can be used to describe the heat transfer process in a dispersed system. It is assumed that the phase transition occurs at a certain temperature $T_f$, so the heat of the "water–ice" phase transition is released only at the boundary of the phase transition. Phase transitions in the seed are not considered, and the seed freezing process will be completed before the average temperature of the seed material reaches $-10\,°C$ (approximate freezing point of water in the bound state).

For the frozen $\Omega^i$ and thawed $\Omega^w$ zones of the dispersed system, as well as the seed material $\Omega^s$, the thermal conductivity equations are applied:

$$
\begin{aligned}
(c\rho)_i \frac{\partial T}{\partial t} &= div\left(\widetilde{\lambda}_i grad\, T\right),\ \vec{x} \in \Omega^i,\ \vec{x} = (x_1, x_2, x_3);\\
(c\rho)_w \frac{\partial T}{\partial t} &= div\left(\widetilde{\lambda}_w\, grad\, T\right),\ \vec{x} \in \Omega^w;\\
(c\rho)_s \frac{\partial T}{\partial t} &= div\left(\widetilde{\lambda}_s\, grad\, T\right),\ \vec{x} \in \Omega^s;\\
\Omega^i(t) &= \left\{\vec{x}\,\middle|\, \vec{x} \in \Omega,\ T(\vec{x},t) < T_f\right\};\\
\Omega^w(t) &= \left\{\vec{x}\,\middle|\, \vec{x} \in \Omega,\ T(\vec{x},t) > T_f\right\};\\
\Omega^i(t) &\cup \Omega^w(t) \cup \Omega^s(t) = \Omega.
\end{aligned}
\tag{1}
$$

where the following obtains: $T$, K—temperature; $t$, s—time; $x_i$, m, $i = 1, 2, 3$—Cartesian coordinates; $c$, J (kg K)$^{-1}$—heat capacity; $\rho$, kg m$^{-3}$—density; $\widetilde{\lambda}$, W (m K)$^{-1}$—coefficient of thermal conductivity.

The calculated zone $\Omega$ consists of a fixed zone $\Omega^s$ ($B_S$ boundary), and two zones $\Omega^i$ and $\Omega^w$ with a variable boundary $B_W$ between them (see Figure 1a).

The law of displacement of the phase transition boundary—the Stefan condition—is determined from the heat balance equation [32]:

$$
\left(\widetilde{\lambda}_i grad\, T - \widetilde{\lambda}_w\, grad\, T,\ grad\, \Phi\right) = L\rho_w \frac{\partial \Phi}{\partial t},
\tag{2}
$$

where $\Phi(\vec{x},t) = 0$ is the phase transition boundary equation; and $L$, J kg$^{-1}$ is the—specific heat of the phase transition.

The temperature at the boundary of the phase transition is continuous:

$$
T(\vec{x} - 0, t) = T(\vec{x} + 0, t) = T_f,\ \ x \in B_w
\tag{3}
$$

where $T_f$, K is the—temperature of the phase transition of water–ice.

To calculate the products of the values of heat capacity $c$ and density $\rho$ in Equations (1), the mixture rule is used:

$$
\begin{aligned}
(c\rho)_i &= mc_i\rho_i + (1-m)c_d\rho_d;\\
(c\rho)_w &= mc_w\rho_w + (1-m)c_d\rho_d;\\
(c\rho)_s &= \varepsilon_1(c_1\rho_1) + \varepsilon_2(c_2\rho_2) + \varepsilon_3(c_3\rho_3),\ \ \varepsilon_1 + \varepsilon_2 + \varepsilon_3 = 1.
\end{aligned}
\tag{4}
$$

Here $m$ is the porosity of a dispersed system based on sand; $\varepsilon_1, \varepsilon_2, \varepsilon_3$ are the volume content of air (index 1), water (index 2) and solid phase (index 3) in the seed. The index d refers to the dispersed phase material (sand).

Similar ratios are used for the thermal conductivity coefficients in the areas under consideration:

$$\widetilde{\lambda}_i = m\lambda_i + (1-m)\lambda_d;$$
$$\widetilde{\lambda}_w = m\lambda_w + (1-m)\lambda_d; \tag{5}$$
$$\widetilde{\lambda}_s = \varepsilon_1\lambda_1 + \varepsilon_2\lambda_2 + \varepsilon_3\lambda_3.$$

To close the thermal part of the sample freezing problem, set the initial condition

$$T(\vec{x}, 0) = T_0, \ \vec{x} \in \Omega, \tag{6}$$

The condition of ideal thermal contact on the internal boundary $B_s$, as well as the boundary condition on the external surface $B$

$$-\widetilde{\lambda}_i \frac{\partial T}{\partial n}\bigg|_B = \alpha(T - T_c), \ \vec{x} \in B, \tag{7}$$

where the following obtains: $\alpha$, W (m$^2$ K)$^{-1}$—heat transfer coefficient; $T_c$, K—temperature of the cooling agent; $\vec{n}$—vector of the unit external normal to the boundary $B$.

### 3.2. Diffusion Task Definition

There is no mass transfer in the frozen zone $\Omega^i$ due to the absence of liquid and gaseous phases according to assumption (I). In the unfrozen zone of «sand-water» dispersion, moisture transport is associated with its absorption by the seed.

Due to simplifying assumptions (I) and (II), the equations of moisture movement in the selected zones have the form [30]:

$$\frac{du}{dt} = 0, \ \vec{x} \in \Omega^i;$$
$$\frac{\partial u}{\partial t} = div(k^w grad\ u), \ \vec{x} \in \Omega^w; \tag{8}$$
$$\frac{\partial u}{\partial t} = div(k^s grad\ u), \ \vec{x} \in \Omega^s.$$

where $u$, kg kg$^{-1}$—moisture content, and $k$, m$^2$ s$^{-1}$—coefficient of hydraulic conductivity.

In the $\Omega^i$-zone $u = u_i = const$, and in the $\Omega^w$- and $\Omega^s$-zones where moisture diffusion is carried out, the rate of diffusion transfer is determined by the coefficients of moisture conductivity $k^w$ and $k^s$.

The boundary conditions of the given diffusion problem have the form:

$$u(\vec{x}, 0) = u_{0w}, \ \vec{x} \in \Omega^i \cup \Omega^w;$$
$$u(\vec{x}, 0) = u_{0s}, \ \vec{x} \in \Omega^s;$$
$$u(\vec{x}, t) = u_{0w}, \ \vec{x} \in B_w; \tag{9}$$
$$-k^w \frac{\partial u}{\partial n}\bigg|_{B_s} = \alpha_D(u_w - u_s), \ \vec{x} \in B_s.$$

where $\vec{n}$—vector of the unit external normal to the boundary $B_s$, and $\alpha_D$, m s$^{-1}$—moisture transfer coefficient.

The formulated mathematical model of heat-and-mass transfer (1–9) when freezing a capsule containing a saturated porous medium and a seed of a forest crop with an internal mobile border is nonlinear. Its research can be carried out using numerical methods.

## 4. Numerical Modelling

A form for a seeding capsule filled with water-saturated dispersed medium, in which the seed is immersed, ready for sowing, is placed in the volume of the chamber with a cooling agent, which acts as carbon dioxide granules. The calculated values of the thermophysical parameters of all components for the studied model are shown in Table 1.

**Table 1.** Calculated parameters for numerical experiment of freezing a single pine seed in a water-saturated dispersed medium.

| Conventional Letter | Name | Means | Unit |
|---|---|---|---|
| $\lambda_w, \lambda_2$ | Thermal conductivity coefficient of water | 0.56 | W (m K)$^{-1}$ |
| $\lambda_i$ | Thermal conductivity coefficient of ice | 2.26 | W (m K)$^{-1}$ |
| $\lambda_d$ | Thermal conductivity coefficient of silica | 8 | W (m K)$^{-1}$ |
| $\lambda_2$ | Thermal conductivity coefficient of air | 0.034 | W (m K)$^{-1}$ |
| $\lambda_3$ | Thermal conductivity coefficient of solid phase of the single seed | 0.04 | W (m K)$^{-1}$ |
| $\rho_w, \rho_2$ | Water density | 1000 | kg m$^{-3}$ |
| $\rho_i$ | Ice density | 920 | kg m$^{-3}$ |
| $\rho_d$ | Silica density | 2650 | kg m$^{-3}$ |
| $\rho_1$ | Air density | 1.225 | kg m$^{-3}$ |
| $\rho_3$ | Seed solid phase density | 1500 | kg m$^{-3}$ |
| $c_w, c_2$ | Water heat capacity | 4212 | J (kg K)$^{-1}$ |
| $c_i$ | Ice heat capacity | 1970 | J (kg K)$^{-1}$ |
| $c_d$ | Silica heat capacity | 750 | J (kg K)$^{-1}$ |
| $c_1$ | Air heat capacity | 1000 | J (kg K)$^{-1}$ |
| $c_3$ | Seed solid phase heat capacity | 2400 | J (kg K)$^{-1}$ |
| $\varepsilon_1$ | Volume content of air in the seed | 0.6 | – |
| $\varepsilon_2$ | Volume content of water in the seed | 0.048 | – |
| $\varepsilon_3$ | Volume content of the solid phase in the seed | 0.352 | – |
| $m$ | Volume content of air in loose dry sand | 0.7 | – |
| $T_c$ | Temperature of the cooling agent | 19,465 (−78,5) [1] | K (°C) |
| $T_0$ | The initial temperature of the sample | 29,315 (+20) | K (°C) |
| $L$ | Specific heat of phase transition «water–ice» | $33.3 \times 10^4$ | J kg$^{-1}$ |
| $k^s$ | Coefficient of moisture conductivity in the seed | $1 \times 10^{-10}$ | m$^2$ s$^{-1}$ |
| $k^w$ | The coefficient of hydraulic conductivity in a sand dispersion | $1 \times 10^{-7}$ | m$^2$ s$^{-1}$ |
| $\alpha_D$ | Moisture transfer coefficient | $2 \times 10^{-6}$ | m s$^{-1}$ |
| $\alpha$ | Heat transfer coefficient «solid $CO_2$—dispersion» | 100 | W (m$^2$ K)$^{-1}$ |
| $a$ | Seed length | 4 | mm |
| $b$ | Seed thickness | 1.5 | mm |

[1] For the production of seed capsules in the field, it is planned to use carbon dioxide granules in the solid phase as a cooler.

It should be noted that there are parameters that should be obtained experimentally, but until now such studies have not been conducted. These parameters include the thermophysical characteristics of the solid phase of the seed, as well as the coefficient of moisture conductivity $k^s$, moisture transfer $\alpha_D$. For model calculations, these values are selected close to the corresponding values for wood matter, as well as for proportionate seeds of agricultural plants. The coefficient of heat transfer $\alpha$ from the carbon dioxide granules to the dispersed mixture in the capsule depends on many factors, including the contact area of the granules with the capsule wall and the intensity of heat transfer. According to the results of the study [33] performed for soil freezing, it is possible to take $\alpha$ from the range of 25–100 W (m$^2$ K)$^{-1}$. The geometric parameters of the seed capsule and seed are illustrated by Figure 1b.

Numerical calculations have shown that the temperature field in the system under study is quite heterogeneous (see Figure 2), which is due to the shape of the capsule and the difference in the thermophysical properties of the materials used in the sample. Figure 2 shows that the selected freezing mode after 300 s leads to the formation of an icy structure in the entire volume of the capsule. The temperature in the seed is uniform and has a value of ~−10 °C, which means no ice crystals in the seed, because the seed moisture content of the seed ($u_{0s}$ = 0.09) corresponds to the presence of only bound water, the freezing point of which is slightly lower than −10 °C.

Simultaneously with cooling through the porous shell, moisture exchange between the seed and the surrounding water dispersion occurs, as well as water transport to the center of the seed. As can be seen from Figure 3, during the period under consideration, the filtration front in the seed remains sufficiently distant from the center, which confirms the possibility of a slight humidity effect on the embryo in the proposed method of preparing seeds for aerial sowing.

Figure 4 illustrates changes in temperature and moisture content over time at certain points on the seed surface. Due to the small size of the seed, the temperature values on its surface differ slightly.

When the water in the dispersion near the seed borders freezes at T = 273 K (approximately t = 180 s), the water supply to the seed stops, and the change in moisture content is possible only due to internal mass transfer.

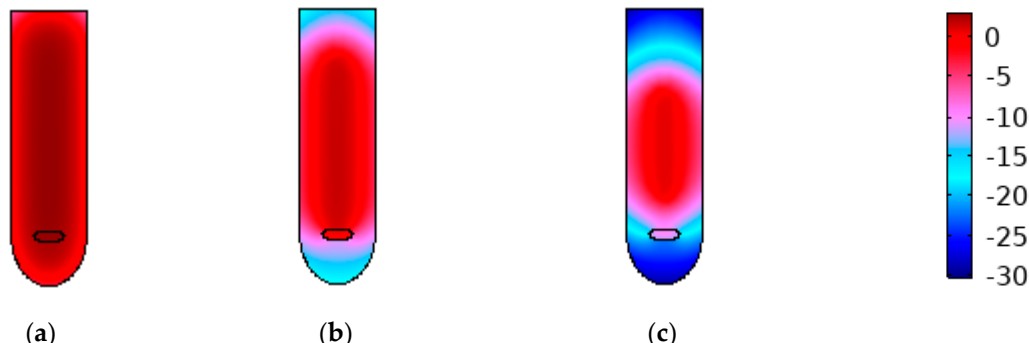

(**a**)　　　　　　　(**b**)　　　　　　　(**c**)

**Figure 2.** Temperature distribution (°C) in the water dispersion of sand–seed system when freezing the seed capsule at times 120 s (**a**), 240 s (**b**), 300 s (**c**) for a seed with a length of 4 mm, a thickness of 1.5 mm.

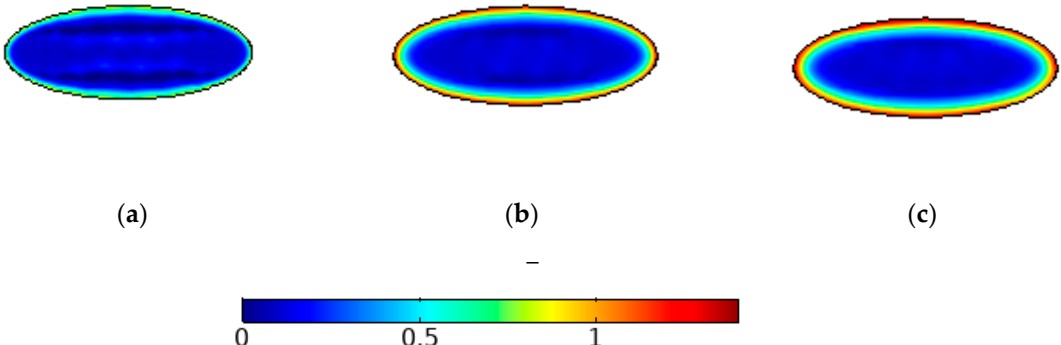

(**a**)　　　　　　　(**b**)　　　　　　　(**c**)

**Figure 3.** Distribution of moisture content (kg kg$^{-1}$) in the seed when the seed capsule is frozen at times 30 s (**a**), 120 s (**b**), 240 s, 300 s (**c**) for a seed with length of 4 mm, and thickness of 1.5 mm.

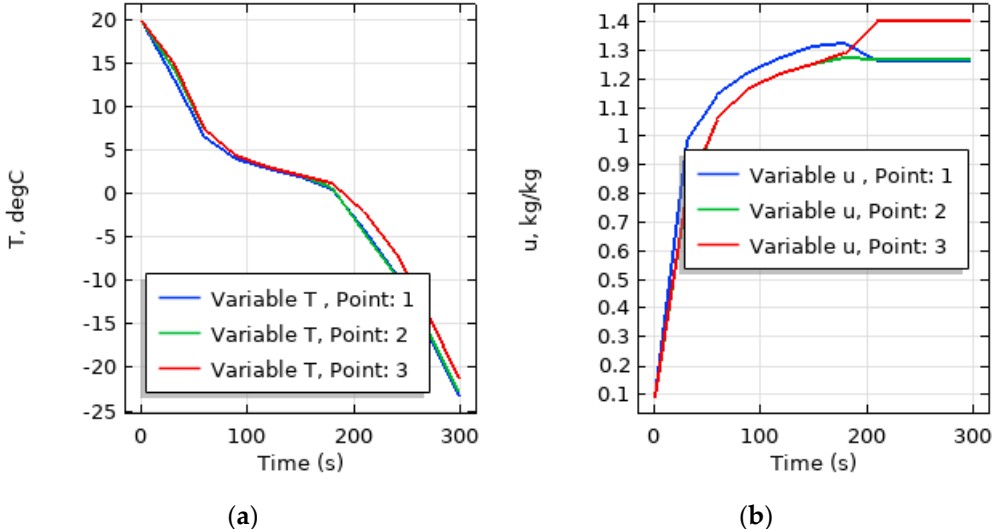

(**a**)　　　　　　　(**b**)

**Figure 4.** Changes over time in the values of temperature (**a**) and moisture content (**b**) on the seed surface at points 1, 2 and 3 marked in Figure 1b for a seed with a length of 4 mm and a thickness of 1.5 mm.

Since the size of the seed is much smaller than the capsules, the aerodynamic resistance is determined by the shape of the capsule and the location of the seed, and the center of pressure must

be below the center of mass of the capsule. The morphometric characteristics of the seed will not significantly affect the accuracy of aerial seeding. The strength of the seed capsule is provided by the presence of a dispersed medium in the form of sand, and the possibility of its production in the field without changing the initial seed germination potential will require from 150 to 250 s.

This study applies only to the treatment of Scots pine seeds before aerial sowing. The feasibility of using this technology for seed storage can be evaluated separately.

## 5. Conclusions

When modeling the process of freezing a capsule containing an aqueous dispersion of sand, in which the seed is immersed, the inhomogeneity of the temperature field in the dispersed medium is established. In contrast, the seed temperature is uniform (about –10 °C). The dynamics of the moisture content distribution in most of the seed volume is 0–0.5 kg kg$^{-1}$. Calculations based on the model allow you to select a range of freezing times for the seed capsule from 150 to 250 s, which provides minimal humidity and temperature effects on seeds that do not reduce the germination rate, as well as securing the requirements for aerial seeding facilities.

**Author Contributions:** Conceptualization, A.N., and O.D.; methodology O.D.; validation, A.N.; formal analysis, A.N. and O.D.; investigation A.N.; data curation, A.N., and O.D.; writing—original draft preparation A.N., and O.D.; writing—review and editing, A.N., and O.D.; visualization, A.N. and O.D. All authors have read and agreed to the published version of the manuscript.

**Funding:** This research received no external funding.

**Conflicts of Interest:** The authors declare no conflict of interest.

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
