# Peer review of "Immersion Freezing of a Scots Pine Single Seed in a Water-Saturated Dispersion Medium: Mathematical Modelling"

_inventions, doi:10.3390/inventions5040051_

Round 1

Reviewer 1 Report

Original article on an innovative method of seed processing. The authors of the unconventional approach to the subject should be congratulated. The study requires a few corrections and additions:

(1) In the introduction, there is no reference to other seed treatment methods that affect, inter alia, ballistic properties. It is about, for example, seed coating or incrustation.

(2) The article absolutely lacks a discussion of the results. Perhaps it is worth recalling the experience of researchers dealing with long-term storage of seed deposits in forest gene banks?

(3) The content of line 179 is a repetition of the content of lines 93-94.

(4) The unit is not given in figure 3.

(5) It should be explained why the dimensions of the seeds were adopted in the mathematical model: length 4 mm, thickness 1.5 mm, since the dimensions of the tested seeds were 4-6 mm and 2-2.5 mm, respectively?

Author Response

Original article on an innovative method of seed processing. The authors of the unconventional approach to the subject should be congratulated. The study requires a few corrections and additions.

The authors sincerely thank the reviewer for his highly professional comments, which significantly contributed to the improvement of the manuscript. We provided the suggested additions in the revised version (marked yellow).

Point 1: In the introduction, there is no reference to other seed treatment methods that affect, inter alia, ballistic properties. It is about, for example, seed coating or incrustation.

Response 1: According to the reviewer's recommendation, the introduction contains references to other methods that change the physical and mechanical properties of seeds, including ballistic ones. However, since the main purpose of the study is immersion freezing of seeds for subsequent aerial seeding, studies related to freezing are mainly presented (see L50-52).

Point 2: The article absolutely lacks a discussion of the results. Perhaps it is worth recalling the experience of researchers dealing with long-term storage of seed deposits in forest gene banks?

Response 2: The authors were unable to find studies related to the freezing of Scots pine seeds in a water-saturated dispersed medium when performing an information search. The research that the reviewer has in mind is related to the" air " freezing of seeds using liquid nitrogen. The authors do not consider it appropriate to compare these processes, however, following the reviewer's recommendation, they indicated that @This study applies only to the treatment of Scots pine seeds before aerial sowing. The feasibility of using this technology for seed storage can be evaluated separately@ (see L217-218).

Point 3: The content of line 179 is a repetition of the content of lines 93-94.

Response 3: We thank the reviewer for the comment. The sentence «The numerical study of the mathematical model (1-9) is based on the COMSOL Multiphysics platform» (previously L179-180), that duplicates the meaning of the sentence on the lines 93-94 (now L96-97) has been deleted.

Point 4: The unit is not given in figure 3.

Response 4: We thank the reviewer for the comment. In figure 3, a unit of measurement for moisture content (kg kg-1) has been added.

Point 5: it should be explained why the dimensions of the seeds were adopted in the mathematical model: length 4 mm, thickness 1.5 mm, since the dimensions of the tested seeds were 4-6 mm and 2-2.5 mm, respectively?

Response 5: In the mathematical model, the seed sizes corresponding to the minimum threshold were adopted. There was a typo in line 87 (now L90). The thickness of Scots pine seed samples is 1.5-2 mm.

Reviewer 2 Report

The authors of this article took up a very interesting, original topic related to modeling of the duration of the seed freezing process in a capsule with a water-saturated dispersed medium. The main approach to solving the problem is based on the patented solution of creating ice capsules for aerial seedings, implemented with the participation of the authors.

The reviewer has no criticism of the methodological approach to the problem, the presented model, and its verification. However, I propose to expand the Introduction chapter with matters related generally to aerial seedings. Of the 31 literature items cited, about half are from Russian authors. I propose to study the literature related to the topic from other regions of the world along with the extension of citations. There is also no general summary of the problem in the form of a separate Discussion chapter. This will increase the quality of the article, making it possible to compare the authors' achievements with the approach of other researchers. Conclusions should relate directly to what the authors have achieved, without general statements.

Detailed comments:

  1. By two authors there is no need to put a comma before and.
  2. The authors have the same affiliation, so there is no need to number it next to the names. It is enough to provide e-mails after affiliation, possibly with initials of them in brackets.
  3. The abstract should be written continuously without dividing into parts, like Research Highlights, etc.
  4. Lines 34-39. This text should be bulleted or written in one sentence. These issues should be expanded and discussed in the Discussion chapter.
  5. Line 57. Why is cryoscarification written in brackets with a question mark?
  6. Line 136. Text after where move the line down.
  7. Line 180. Write Form with a lowercase letter.
  8. Line 218 and others. All text should be written impersonally.
  9. Line 257. The year 2020 occurs twice at the end.

Author Response

The authors of this article took up a very interesting, original topic related to modelling of the duration of the seed freezing process in a capsule with a water-saturated dispersed medium. The main approach to solving the problem is based on the patented solution of creating ice capsules for aerial seedings, implemented with the participation of the authors.

We thank the reviewer for the valuable comments and provided the suggested additions in the revised version (marked yellow).

Point 1: The reviewer has no criticism of the methodological approach to the problem, the presented model, and its verification. However, I propose to expand the Introduction chapter with matters related generally to aerial seedings. Of the 31 literature items cited, about half are from Russian authors. I propose to study the literature related to the topic from other regions of the world along with the extension of citations. There is also no general summary of the problem in the form of a separate Discussion chapter. This will increase the quality of the article, making it possible to compare the authors' achievements with the approach of other researchers. Conclusions should relate directly to what the authors have achieved, without general statements.

Response 1: Aerial seeding in world practice is quite widely and relatively recently discussed in the article [1] with the participation of one of the co-authors. The main requirements formed on the basis of these data for seeds are presented in the bulleted list (see L34-41). The main purpose of the article is to model and justify the time of possible immersion freezing of seeds for aerial seeding, which is shown in the results section. These provisions are discussed in the results and discussion section (see L217-224). Conclusions clarified (see L226-232).

Point 2: By two authors there is no need to put a comma before and.

Response 2: The comma before "and" is removed (see L5)

Point 3: The authors have the same affiliation, so there is no need to number it next to the names. It is enough to provide e-mails after affiliation, possibly with initials of them in brackets.

Response 3: Adjusted in accordance with referee recommendations (see L7-8)

Point 4: The abstract should be written continuously without dividing into parts, like Research Highlights, etc.

Response 4: In the abstract, on the recommendation of the reviewer, subheadings were removed (see L 11-27)

Point 5: Lines 34-39. This text should be bulleted or written in one sentence. These issues should be expanded and discussed in the Discussion chapter.

Response 5: The text is labeled according to the log template. These issues are discussed in the relevant section (see L217-222).

Point 6: Line 57. Why is cryoscarification written in brackets with a question mark?

Response 6: The wrong question mark was removed (see L60).

Point 7 Line 136. Text after where move the line down.

Response 7: We ask the reviewer to clarify this recommendation. Line 136 (now L139) provides an explanation of the equation parameters in accordance with the article design template.

Point 8 Line 180. Write Form with a lowercase letter.

Response 8: Corrected (see L182).

Point 9. Line 218 and others. All text should be written impersonally.

Response 9: Corrected (see L226-232).

Point 10. Line 257. The year 2020 occurs twice at the end.

Response 10: Corrected (see L266).